# Who is the Spy?

**Chentian Wei**
Institute for Network Sciences and Cyberspace
Tsinghua University
Beijing, China 100084
wct24@mails.tsinghua.edu.cn

**Jiewei Chen**
School of Software
Tsinghua University
Beijing, China 100084
chen-jw24@mails.tsinghua.edu.cn

**Jinzhu Xu**
Department of Computer Science and Technology
Tsinghua University
Beijing, China 100084
xujz23@mails.tsinghua.edu.cn

## Abstract

With the expansion of small scale games and prompt engineering, logical reasoning and problem solving has become integral to our daily lives. Simultaneously, recent advancements in large language models (LLMs) and techniques of prompt engineering have transformed the fields of game development and Artificial Intelligence (AI) through Machine Learning (ML). Given the rapid progress in this research area, there is a pressing need for a comprehensive overview that encapsulates the current state of LLM and prompt engineering. Therefore, we propose the idea of implementing Chain of Thought (COT) prompting to guide the LLM to follow a reasoning process in solving the riddles of a small games. This brings us to the title **"Who is the Spy?"** for our final project.

In this proposal, we first present the foundational background of large language models (LLMs), Prompt Engineering, and Chain of Thought (CoT) prompting in small scale games. Next, we conduct a review of the relevant literature on prompt engineering principles (CoT and ToT), which will be explored further in our final report. Finally, we identify and discuss the propose method in this domain that merit further investigation.

## 1 Background

### 1.1 Large Language Model

With the introduction of large language models like ChatGPT3 and 4 [7], it is clear that LLMs have emerged as a cutting-edge artificial systems that covers the architecture training approach, and a wide range of applications in natural language processing, reasoning, and text generation with coherent communication, with generalization to multiple tasks (Naveed et al., 2024 [5]).

LLMs appear to have emergent abilities, such as reasoning, planning, decision-making, in-context learning, and answering in zero-shot settings, allowing them to be widely popular and diversely adopted throughout different scenarios.

Through the acquisition of vast amounts of data, LLMs have shown the importance of prompting when dealing with complex problems.

38th Conference on Neural Information Processing Systems (NeurIPS 2024).

## 1.2 Prompt Engineering

Prompt engineering is a technique that guide LLMs to follow specific logical steps, by breaking down complex problems into smaller manageable tasks. This approach allows models to provide more accurate and coherent responses across multiple domains.

Sahoo et al., 2024 [8] emphasizes the importance of prompt engineering on improving LLMs logical understanding by using input prompts optimization and contextual information to guide the model using task-specific information. Taking into account that the quality of the input might have a significant impact on the model's output that activate relevant knowledge without modifying the core model parameters.

Chain of Thought (CoT) prompting helps models to think step by step and focusing on how well-designed prompts may enhance LLMs reasoning abilities. COT has proven useful in activities that need more advanced problem-solving abilities, such as complicated QnA, riddles or detecting loopholes in a small game that need to be solved thoroughly.

## 1.3 Chain of Thought (COT) Prompting in a Small Scale Games

CoT prompt in a small games implements multi-agent framework that operates through various intelligent agents reasoning and acting in the language-based board game. Observing the prompts or statements given by the player step by step through logic reasoning and decision making. By breaking down complex scenarios, each agent dynamically evaluate the game's context, predict potential outcomes, and adjust strategies. With the help of CoT prompting, agents mimic human-like reasoning avoiding biases.

In the context of one of Chinese games called "shei shi wo di" or "Who is the spy" in English, all the agents are assigned one of two different but similar keywords, dividing them into two different camps: villager and spy. CoT plays role in both deception (spy) and detection (villager) strategies. In the deception side, CoT simulates logical replies, maintains consistency, and predicts others players reaction, misleading clues to other players. In advance, the step-by-step logic ensures the wolf's story matches the expectations of the other participants, making their identity more difficult to find. On the other hand, CoT identify logical loopholes by breaking down each statement given by other players systematically, finding out inconsistencies or contradictions of players' responses to find the highest possibility of the player who plays role as the spy. In addition, probing deeper with strategically framed questions makes it easier for players to expose flaws in the spy's reasoning.

To conclude, CoT handles errors at a scale and ensures that decisions are based on careful analysis and complete reasoning. LLMs process multiple scenarios, processes, and possible outcomes in a fraction of the time, which leads to quicker and more comprehensive analysis. CoT-guided LLMs organize thoughts and reasoning clearly without emotional interference and maintain a consistent, logical approach throughout result in a more reliable decision making. Lastly, CoT prompting perfoms systematic breakdown prevents overlooking critical details, leading to more precise conclusions and ultimately, which result in better game-play outcomes.

## 2 Related Works

LLMs have made significant progress in solving reasoning and planning problems. Researchers have widely applied these models to tasks like board games and social deduction games, achieving remarkable success. In board games, Noever et al.[6] fine-tuned the GPT-2 model using Portable Game Notation (PGN), optimizing 774 million parameters. This enabled the model to generate reasonable strategies and exhibit game patterns recognizable as classic openings. ChessGPT[1] combines strategy learning and language modeling, utilizing a large dataset of chess games to enhance the model's ability to solve complex chess positions. In social deduction games, Xu et al.[13] proposed a framework that allows LLMs to participate in these games without fine-tuning. By reviewing historical dialogues, the model improves its reasoning abilities, making it applicable to social deduction games that rely on natural language interaction. Xu et al.[14] also developed strategic language agents using reinforcement learning (RL) combined with LLMs for games like "Werewolf." These agents can generate diverse actions during the game and choose the best option from multiple candidates, achieving near-human-level strategic play. Wu et al.[11] introduced the "Thinker" module, which processes player speech with structured analysis and deep logical evaluation, enhancing the

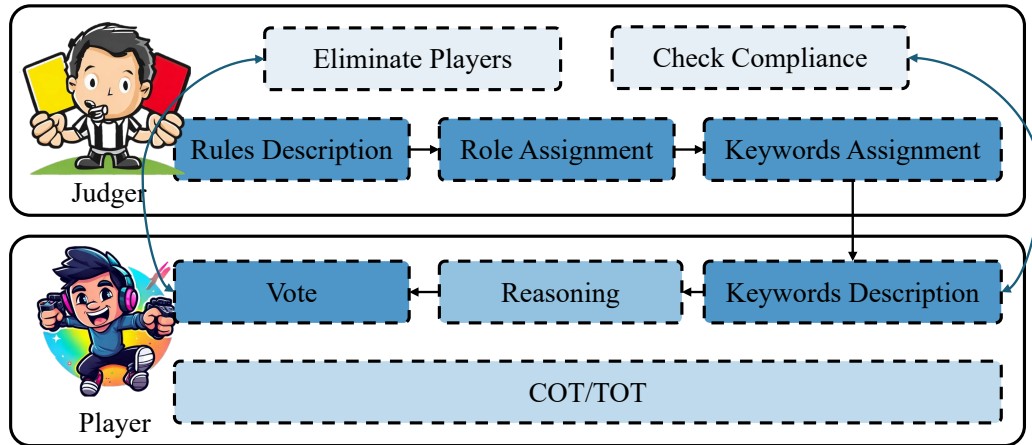

Figure 1: Our Framework

reasoning and decision-making capabilities of LLMs in games like "Werewolf," where complex language reasoning is required.

Studies (e.g., [2, 12]) have shown that combining LLMs with world models enhances their planning and reasoning capabilities by generating action plans and predicting future states, leading to strong performance in tasks like logical reasoning, math, and coding. Ku et al.[9] explored how multi-agent discussion frameworks enhance LLM reasoning, finding that even single-agent setups can achieve comparable results in collaborative reasoning tasks through strong prompting. CR-Planner[4] combines retrieval-augmented generation and critic models to improve reasoning and planning abilities.

In recent years, Chain of Thought (CoT)[10] and Tree of Thought (ToT)[15] have achieved significant success in enhancing LLM reasoning and planning capabilities. Kim et al.[3] demonstrated how CoT fine-tuning improves LLM performance in few-shot learning tasks, significantly boosting zero-shot and few-shot learning abilities in reasoning tasks by fine-tuning CoT datasets. Zhang et al.[16] combined CoT and ToT with preference optimization to enhance reasoning. The method generates multiple candidate paths during reasoning and evaluates each step, optimizing the entire reasoning chain and further improving reasoning quality. The research aims to leverage CoT and ToT to improve LLM performance in reasoning tasks, ultimately enhancing LLM reasoning capabilities in social deduction games like "spy."

## 3 Definition and Method

### Game Definition: "Who is the spy"

"Who is the spy" is a highly popular party game in China where players must use dialogue and reasoning to identify the spy or antagonist character hidden within the group. The core gameplay of this title shares similarities with "Werewolf" and "Mafia" as they all revolve around hidden identities, deception, and deduction.

The process of the game is as follows:

1. Rule Description: Explain the game rules to the large language model, including keyword description, players elimination, and voting.

2. Role Assignment: Assign each player an initial role, which could be the spy or a villager. Ensure the model is aware of its assigned role.

3. Keyword Assignment: Assign a keyword to each role in the game. One is the spy's keyword, while the others are similar but not identical to the spy's keyword. Inform the large language model of the keyword it has been assigned.

4. Keywords Description : Players take turns describing the keyword they have received, and the large language model describes the keyword based on the given information.

5. Vote: At the end of this round, all players eliminate the non-spy and vote to select the player that has the highest possible as the spy role.

Based on the structure described above, as shown in the figure 1, we divide the overall framework into two parts: the referee and the players.

In the game, the referee will control the progression of the game. First, the referee will explain the game rules and assign roles to all players, as well as distribute key terms. During the game session, the referee will continuously check whether the players' outputs comply with the rules. If an output does not adhere to the game rules, the player will be required to output again, and the type of violation will be recorded. Next, the referee will compile all descriptions by role and distribute them uniformly to the players for reasoning and voting. The referee is responsible for collecting the voting results and carrying out the elimination process. In the game, both large language models and humans may participate simultaneously, and they will play according to the game rules.

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
