# OpenReview forum: "Who is the Spy?"
_tsinghua.edu.cn/THU/2024/Fall/AML — THU 2024 Fall AML Submission_

### Official Review · ~Thomas_Adler2 · 2024-11-05
**Clear explanation of the problem but not of the methods**

**Rating:** 9
**Confidence:** 4

**Review:**

The explanation of CoT, LLMs and the game was very clear. It seems like it is an idea pushing the frontiers of LLMs. The related work sections shows the possibilities and potential of this idea working. However, the proposal goes above the 2 page limit, but more importantly there is a lack of specificity about the method that will be employed. Would have liked to see more of the proposal spent on the specifics of the implementation, even if not definite.

---

### Official Review · ~Yinuo_Li1 · 2024-11-06
**Interesting topic but no clear statement of approach method**

**Rating:** 7
**Confidence:** 4

**Review:**

This proposal has a very clear and interesting topic, and gives a solid review of relevant concepts including LLM, prompt engineering, and CoT prompting in games. This proposal also has a very clear defination about their base game who is the spy.

However, I couldn't find their main purpose except " we propose the idea of implementing Chain of Thought (COT) prompting to guide the LLM to follow a reasoning process in solving the riddles of a small games" in the abstract. The proposal does not clearly show the value of doing this and the main challenge other people are facing on this problem, therefore, I can't clearly understant the significance of this certain problem and their job just by reading the proposal.

---

### Official Review · ~Yanchen_Wu1 · 2024-11-08
**Interesting research**

**Rating:** 8
**Confidence:** 4

**Review:**

This work tells an interesting story: using large models to check for holes in the logic of a player's statement and identify them. This is a very difficult task, so how to train this model well? There is no clear method in the proposal. I look forward to further research. And the proposal exceeded the page limit.

---

### Official Review · ~Renrui_Tian1 · 2024-11-09
**Clear Problem Definition and Strong Background, but Methodology and Contributions Need More Specificity**

**Rating:** 7
**Confidence:** 4

**Review:**

**Strengths**:
* **Clear Problem Definition**: The proposal effectively defines the problem of using Chain of Thought (CoT) prompting to guide Large Language Models (LLMs) in solving riddles within a game setting.
* **Relevant Background**: The proposal provides a comprehensive background and a relevant literature review of existing research on LLMs for gaming, demonstrating a strong understanding of the field.
* **Game Framework**: The proposal outlines a clear framework for implementing the game "Who is the Spy?" with both human and LLM participants, including the roles of the referee and players.

**Areas for Improvement**:
* **Specificity of Methodology**: The proposal could benefit from a more detailed description of the specific CoT prompting techniques that will be used.
* **Challenges and Contributions**: The proposal falls short in explaining the potential challenges and anticipated contributions of the proposed method.
* **Page Limit Compliance**: The proposal exceeds the specified page limit, and a more concise presentation would help improve clarity and adherence to requirements. The Background and Related Work sections could be more concise to avoid excessive use of pages.

**Overall**:
The proposal presents a well-defined research problem and demonstrates a strong understanding of the relevant background and existing literature. The proposed method has the potential to contribute to the advancement of AI in social deduction games. However, further details on the methodology, potential challenges, anticipated contributions, and adherence to page limitations are needed to strengthen the proposal.

---

### Official Review · ~Chenxi_Hu4 · 2024-11-11
**Novel Application of CoT in Games, but Limited Implementation, Evaluation and Challenges**

**Rating:** 8
**Confidence:** 4

**Review:**

The proposal presents a novel application of CoT prompting in “Who is the Spy?” with clear objectives and a solid background. However, it requires further detail on CoT implementation, evaluation metrics, and potential challenges. Addressing these aspects would enhance the project’s feasibility.

---

### Official Review · ~Zhixuan_Pan1 · 2024-11-11

**Rating:** 6
**Confidence:** 4

**Review:**

The project aims to use Chain of Thought (CoT) prompting to enhance reasoning in the game “Who is the Spy?”.

Pros:

1. They design an interesting game scenario.

2. The background and related work have been thoroughly reviewed.

Cons:

1. If this work remains solely at the level of prompt engineering, I do not believe it has significance beyond that. It seems like an interesting game application rather than a meaningful research work.

2. A quantitative evaluation metric is lacking to assess how well the model performs in the game.

---

### Official Review · ~Mingdao_Liu1 · 2024-11-11
**Review for "Who is the Spy?"**

**Rating:** 7
**Confidence:** 3

**Review:**

The proposal aims to implement CoT prompting to guide the LLM to follow a reasoning process in solving the riddles of a small game, specifically "Who is the Spy".

Pros:
1. The proposal includes all required parts for a proposal.
2. The proposal is easy to follow and includes a comprehensive review of the background, related works, and game rules.

Cons:
1. The proposal does not include the research contribution or a detailed plan. Applying CoT to a game might not be meaningful enough for a research project.
2. (minor) The proposal is more than 2 pages.

---

### Official Review · ~Jiajun_Xu3 · 2024-11-11
**Interesting topic but lack of deep analysis and detailed methodologies**

**Rating:** 7
**Confidence:** 4

**Review:**

The proposal provides an interesting perspective on using Chain of Thought (CoT) prompting to enhance large language model (LLM) performance in the game "Who is the Spy?", a social deduction game.
The topic is pretty novel and the proposal presents solid background information. However, the significance of this work seems yet to be confirmed. And the document lacks detailed methodologies on how CoT will be implemented technically in LLMs beyond general descriptions of game phases.
Furthermore, it has exceeded the page limit in the guidelines.

---

### Official Review · ~XueZeng1 · 2024-11-11

**Rating:** 7
**Confidence:** 3

**Review:**

The proposal selects popular party game "Who Is the Spy" in China as the research background is quite suitable.

Pros:

1.Clear design of the game process and clear division of the game framework.
2.This design of mixed participation is innovative and of practical significance.

Cons:
Some more in-depth considerations are required for this proposal.There is no detailed coping strategy for other possible abnormal situations, such as a player dropping out midway.

---

### Official Review · ~KAI_JUN_TEH1 · 2024-11-11
**Interesting games**

**Rating:** 9
**Confidence:** 4

**Review:**

The article provides a very detailed definition and explanation of "Who is a spy," which is commendable. However, most of the proposal mainly focuses on explaining the issues/games and related work. I would like to learn more about the authors' thoughts.

---

### Official Review · ~Wanlan_Ren1 · 2024-11-12
**Review for "Who is the Spy?"**

**Rating:** 7
**Confidence:** 4

**Review:**

The proposal offers an ambitious exploration of applying Chain of Thought (CoT) prompting within large language models (LLMs) to enhance reasoning capabilities in a popular Chinese game, "Who is the Spy?" The proposal is intriguing, addressing the challenge of logical reasoning in social deduction games by leveraging CoT prompting. Strengths include a thorough background on LLMs, prompt engineering, and related work, setting a strong foundation for understanding the complexities of the game mechanics and LLM integration. However, clarity could be improved in some sections, particularly regarding CoT's role in deception versus detection strategies. Additionally, more concrete experimental plans and success metrics would strengthen the impact.